# The Status of South Africa's Freshwater Fish Fauna: A Spatial Analysis of Diversity, Threat, Invasion, and Protection

Mohammed Kajee [1,2,3,*] , Helen F. Dallas [2,4], Charles L. Griffiths [1], Cornelius J. Kleynhans [5] and Jeremy M. Shelton [2]

1 Department of Biological Sciences, University of Cape Town, Cape Town 7700, South Africa; charles.griffiths@uct.ac.za
2 Freshwater Research Centre, Cape Town 7975, South Africa; helen@frcsa.org.za (H.F.D.); jeremy@frcsa.org.za (J.M.S.)
3 DSI/NRF Research Chair in Inland Fisheries and Freshwater Ecology, South African Institute for Aquatic Biodiversity (SAIAB), Makhanda 6140, South Africa
4 Faculty of Science, Nelson Mandela University, Gqeberha 6031, South Africa
5 Department of Water Affairs and Forestry (Retired), 8th Avenue 771, Wonderboom South, Pretoria 0084, South Africa; kneria@gmail.com
* Correspondence: kjxmoh007@myuct.ac.za

**Abstract:** In South Africa, freshwater habitats are among the most threatened ecosystems, and freshwater fishes are the most threatened species group. Understanding patterns in freshwater fish diversity, threat, invasion, and protection status are vital for their management. However, few studies have undertaken such analyses at ecologically and politically appropriate spatial scales, largely because of limited access to comprehensive biodiversity data sets. Access to freshwater fish data for South Africa has recently improved through the advent of the Freshwater Biodiversity Information System (FBIS). We used occurrence records downloaded from the FBIS to evaluate spatial patterns in distribution, diversity, threat, invasion, and protection status of freshwater fishes in South Africa. Results show that record density varies spatially, at both primary catchment and provincial scales. The diversity of freshwater fishes also varied spatially: native species hotspots were identified at a provincial level in the Limpopo, Mpumalanga, and KwaZulu-Natal provinces; endemic species hotspots were identified in the Western Cape; and threatened species hotspots in the Western Cape, Mpumalanga, Eastern Cape, and KwaZulu-Natal. Non-native species distributions mirrored threatened species hotspots in the Western Cape, Mpumalanga, Eastern Cape, and KwaZulu-Natal. Some 47% of threatened species records fell outside of protected areas, and 38% of non-native species records fell within protected areas. Concerningly, 58% of the distribution ranges of threatened species were invaded by non-native species.

**Keywords:** freshwater fishes; South Africa; biodiversity data; FBIS; species richness; threatened species; occurrence data

**Key Contribution:** This study uses occurrence records downloaded from the FBIS to evaluate spatial patterns in distribution, diversity, threat, invasion, and protection status of freshwater fishes in South Africa. There is an urgent need for better monitoring of freshwater fishes in South Africa, so that large-scale assessments of the status of the country's freshwater fish fauna can be more accurately assessed.

## 1. Introduction

Freshwater ecosystems across the world are under threat because they face major impacts from human activities, including land-use change, non-native invasives, water over-abstraction, and the climate crisis [1–4]. Consequently, many organisms that rely on these habitats are threatened with extinction [5,6]. Recent reports indicate that almost one-in-three freshwater taxa are threatened with extinction globally [7,8]. This is of major

concern, given that freshwater ecosystems account for ~10% of global biodiversity and 51% of all fish species, despite only covering 1% of the earth's surface [8–10].

In South Africa, freshwater habitats (wetlands and rivers) are among the most threatened ecosystem types, with freshwater fishes the country's most threatened species group [11]. Of 105 formally described native species, 25 are classified as threatened (Vulnerable, Endangered, or Critically Endangered) by the International Union for the Conservation of Nature (IUCN) Red List of Threatened Species [12]. For the 40 South African-endemic freshwater fish species, the threat level is even higher, with two-thirds of endemics being currently classified as threatened [13]. It is thus critical that trends in diversity, distribution, threat status, and protection status of freshwater fishes in South Africa are comprehensively assessed to support management approaches and conservation action.

South Africa has a rich history of freshwater fish research dating back more than 200 years [14]. The first comprehensive catalogue of freshwater fishes for southern Africa was written by G.D.F. Gilchrist and W.W. Thompson between 1913 and 1917 [14]. Skelton's [14] field guide—*A Complete Guide to the Freshwater Fishes of Southern Africa*—followed with updated information on species diversity and distribution [14]. Skelton et al. [15] undertook the first broad-scale analysis of distribution, richness, endemism, and conservation status of freshwater fishes in South Africa, Lesotho, and the Kingdom of eSwatini, including 94 native fish taxa and 18 non-native species [15]. In 2011, an atlas of National Freshwater Ecosystem Priority Areas for South Africa (NFEPA) was published, which summarised data and expert knowledge of regional freshwater ecosystems and fish distributions [16].

The most recent national-scale assessment of freshwater fishes in South Africa, by Chakona et al. [13], summarised the diversity, distribution, and extinction risk of native freshwater fishes. They assessed the extinction risk for 101 valid species and 18 unique genetic lineages of native fishes, finding that 36% of South Africa's freshwater fishes are threatened with extinction. Important diversity and threat hotspots were also assessed, with the Cape Fold Ecoregion [17] being identified as both a high-diversity and high-threat region [13].

In addition to peer-reviewed research articles, the South African National Biodiversity Institute (SANBI), which is mandated to assess and monitor the state of South Africa's biodiversity through the National Environmental Management: Biodiversity Act: Act 10 of 2004 [18], also assessed the status of South Africa's freshwater fishes in their National Biodiversity Assessments (NBA) conducted in 2004, 2011, and 2018 [11]. The most recent NBA [11] assessed the threat status of 118 native freshwater fishes and found that freshwater fishes contained the highest percentage of threatened taxa of any species group in the country [11]. Whilst this assessment has been an important update to the status of freshwater fishes in the country, no specific spatio-temporal analyses of the available fish occurrence data were conducted.

Regardless of this well-established research infrastructure, long-term monitoring data sets for South Africa's freshwater fishes are limited [13,19], with no formalised national or even provincial monitoring programmes currently being undertaken. Scott [20], along with Skelton et al. [21], presented the only known Atlas of southern African freshwater fishes, which contained 35,180 georeferenced specimen records covering 735 species from across the region [20,21]. Data from the atlas have since been uploaded to the GBIF and used to evaluate fish distributions and links with environmental gradients at regional scales [21,22]. More recently, however, platforms such as FishBase [23] and GBIF [24] have facilitated the storage of, and access to, large databases via online, open access platforms, which has allowed government and research organisations, as well as private individuals, to share and access data freely online. For example, the South African Institute for Aquatic Biodiversity (SAIAB) uploaded its entire freshwater fish database to the GBIF platform [25], thereby greatly improving access to this data set [26].

Despite the wealth of freshwater research and biodiversity information available in South Africa, there was no central database for housing freshwater biodiversity data until the recent development of the Freshwater Biodiversity Information System (FBIS;

freshwaterbiodiversity.org) [19]. The FBIS is a data-rich, open access online platform that hosts, analyses, and serves freshwater biodiversity data [19], and aims to serve as a platform for the inventory and maintenance of freshwater data, improving access to comprehensive and reliable freshwater biodiversity data [19]. Consequently, the FBIS functions as a repository for freshwater biodiversity data in South Africa and has been populated with data from a variety of key sources, including published scientific literature, government organisations, and online databases [19]—making it the first comprehensive, accessible national-level resource for freshwater biodiversity data in the country [19]. The database currently hosts more than 57,000 occurrence records for freshwater fishes in South Africa.

Given growing anthropogenic pressures on freshwater ecosystems and fish, and recent improvements in freshwater fish data access in South Africa, there is now both an urgent need and new opportunity for a data-driven assessment of trends in diversity, distribution, and threat status of South Africa's freshwater fish fauna to support improved management and conservation decisions. Freshwater fishes in South Africa face pressure from water abstraction [15], climate change [27–30], and the introduction of non-native species [31–33]. Given the threats to, and observed declining trends in, South Africa's freshwater fish fauna, assessing the conservation status and effectiveness of conservation areas at protecting threatened fishes is both imperative and urgent [13,34].

South Africa has a relatively extensive terrestrial protected area network covering an area of approximately 270,000 km$^2$ [35,36]. It includes both statutory conservation areas (e.g., National Parks) and non-statutory conservation areas (e.g., Private Nature Reserves) [35–37]. Whilst these protected areas offer some protection to freshwater fishes, few conservation areas and reserves protect entire catchments [15,38], with freshwater systems, in general, being especially neglected [39]. Consistent with global trends [40–42], South Africa's network of protected areas is severely lacking in terms of adequately conserving freshwater ecosystems, with over 90% of the country's main rivers falling outside protected areas [43,44]. The current network only includes small components of protected river areas that form part of much larger, degraded aquatic systems further upstream and downstream of the parks [15,45]. Recent studies have shown that the current protected area network does not adequately protect native freshwater fish [34,45,46], with 84% of taxa regarded as under-protected [34]. Given that approximately 90% of freshwater species listed as Critically Endangered, Endangered, or Vulnerable on the IUCN Red List [12] are threatened by human-induced habitat loss [44], providing adequate protection and interventions to prevent further habitat loss and degradation is of utmost importance. Incorporating spatial freshwater biodiversity data into both protected area planning and management in South Africa will improve the role of protected areas in conserving freshwater ecosystems [47–50].

We used historic-to-present-day freshwater fish data currently available on the FBIS database to assess spatial patterns of distribution, diversity, invasion, and threat status in South Africa at provincial and primary catchment scales. Additionally, we assessed how well South Africa's protected area network protects threatened species and fish diversity hotspots.

## 2. Materials and Methods

### 2.1. Study Site

The geographic scope of this study was restricted to the Republic of South Africa (Figure 1). Occurrence records were also limited to rivers, dams, and freshwater lakes, with marine systems in South Africa excluded. South Africa's freshwater fish fauna are managed at provincial level (Figure 1B) via individual provincial conservation authorities. However, primary hydrological catchments (Figure 1C) and freshwater ecoregions (Figure 1D) represent more ecologically relevant assessment and management extents, given that the distribution of fishes are strongly impacted by the climate, geomorphological history, and topography of each region [14]. Analyses for this study were therefore conducted at both provincial and primary catchment scales.

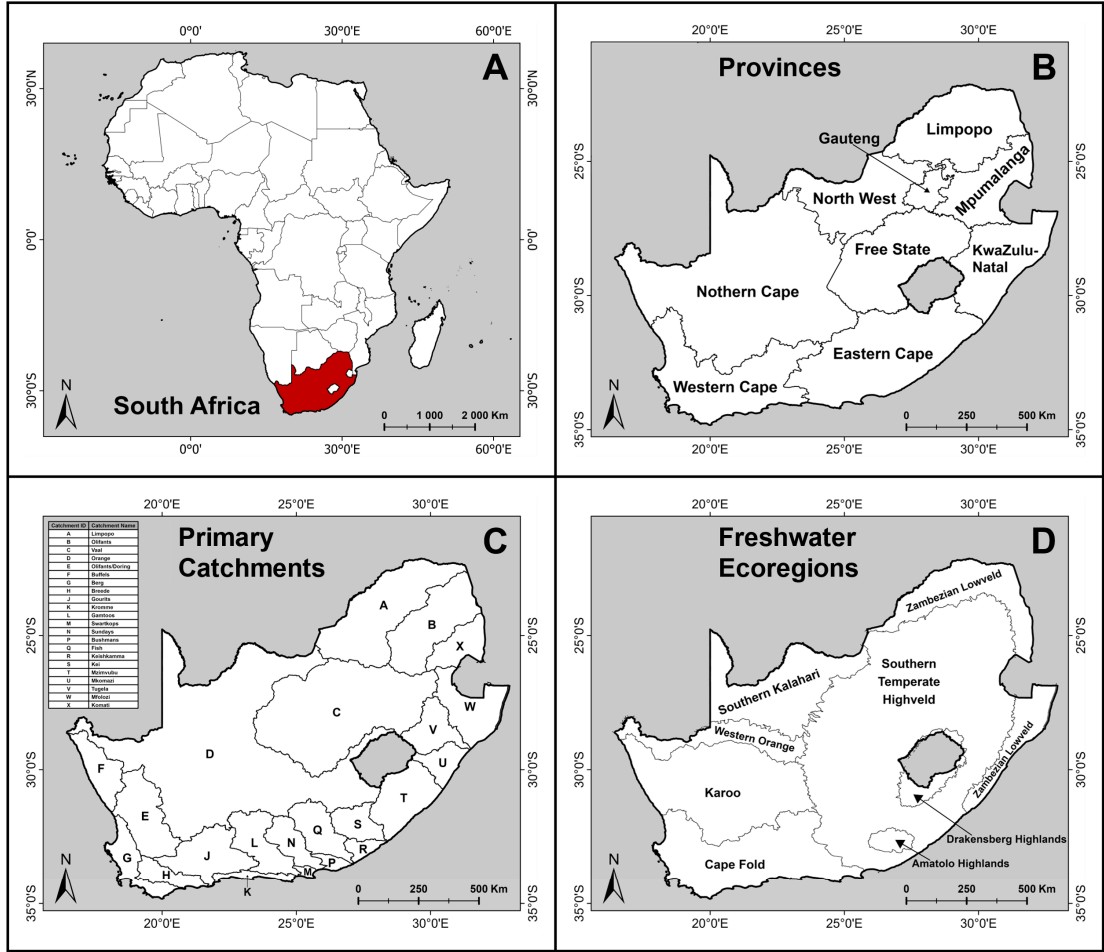

**Figure 1.** Republic of South Africa (**A**) showing provincial boundaries (**B**), primary catchments (**C**), and freshwater ecoregions (**D**). Catchment IDs are represented, where: A = Limpopo; B = Olifants; C = Vaal; D = Orange; E = Olifants/Doring; F = Buffels; G = Berg; H = Breede; J = Gourits; K = Kromme; L = Gamtoos; M = Swartkops; N = Sundays; P = Bushmans; Q = Fish; R = Keishkamma; S = Kei; T = Mzimvubu; U = Mkomazi; V = Tugela; W = Mfolozi; and X = Komati.

*2.2. Data Collection and Cleaning*

Occurrence records for all freshwater fish (all known primary and secondary freshwater fish, as well as catadromous species) occurring in South Africa were downloaded from the Freshwater Biodiversity Information System (FBIS) on 30 July 2023 [51] for further analysis.

Preliminary data cleaning was conducted in R [52]; Version 4.2.3 and ArcGIS Pro [53] as follows. First, the data set was clipped to the political boundary of the Republic of South Africa (Figure 1), ensuring that all records in the ocean were removed from the data set. Next, a list of native species that have been translocated outside of their native range (see Supplementary Material S1) was extracted from Ellender and Weyl [32]. For species known to be native but extralimital [32], records occurring outside of the native range of the species—based on IUCN range maps [54] and expert knowledge—were flagged. These records were not included in the species richness counts for native, threatened, and endemic analyses. The R package, *biogeo* [55] was then used to 'autoclean' the data set prior to final analyses. This was done using the '*quickclean*' function in *biogeo* [56], which performs a check to determine if records are at appropriate spatial resolution, removes records that are deemed to be erroneous, and flags duplicate records per species per grid cell. Supplementary Material S4 provides a list of data sources included in the final data set.

*2.3. Data Analysis*

All data analyses were conducted using R ([52]; Version 4.2.3) and ArcGIS Pro [53]. Using the *richnessmap* function from the R package, *biogeo* [55], we produced species richness maps at a quarter-degree square (QDS) spatial scale (15′ resolution). We used a QDS resolution since accurate distribution maps for all freshwater fishes occurring in South Africa do not currently exist [13]. This spatial scale was also used by Skelton et al. [21], Scott et al. [20], and Skelton et al. [15]. We produced richness maps for all native, non-native, and threatened (listed as Vulnerable, Endangered, or Critically Endangered according to the IUCN Red List of Threatened Species) native species. Species richness was also assessed for species endemic to a single freshwater ecoregion [19,26]. We calculated species counts and the number of records for each species for each province and within each primary catchment, these being considered politically and ecologically useful scales, respectively.

We conducted protected area analyses by overlaying occurrence records and species richness maps at QDS resolution with the protected areas spatial layer downloaded from the South African Protected and Conservation Areas Database on 10 September 2023 [35,36]. The intersections were then used to calculate number of records and distribution area found within protected and conservation areas for native, non-native, threatened, and endemic species. Additionally, we used species richness maps at QDS resolution for threatened and non-native species to assess spatial overlap between these two species groups. Final maps were produced in ArcGIS Pro [53].

## 3. Results

A total of 57,485 records for freshwater fish were downloaded from the FBIS. After data cleaning, these were reduced to 55,215 records, comprising both native (*n* = 50,927) and non-native (*n* = 4288) fishes occurring in South Africa (Figure 2). The final, cleaned data set spans 184 years (1839–2023) and represents 129 species, of which 105 (81%) were native and 24 (19%) were non-native. A list of species with occurrence records in South Africa is available in Kajee et al. [26]. The data vary in time, with 89% of records being collected between 1975 and present.

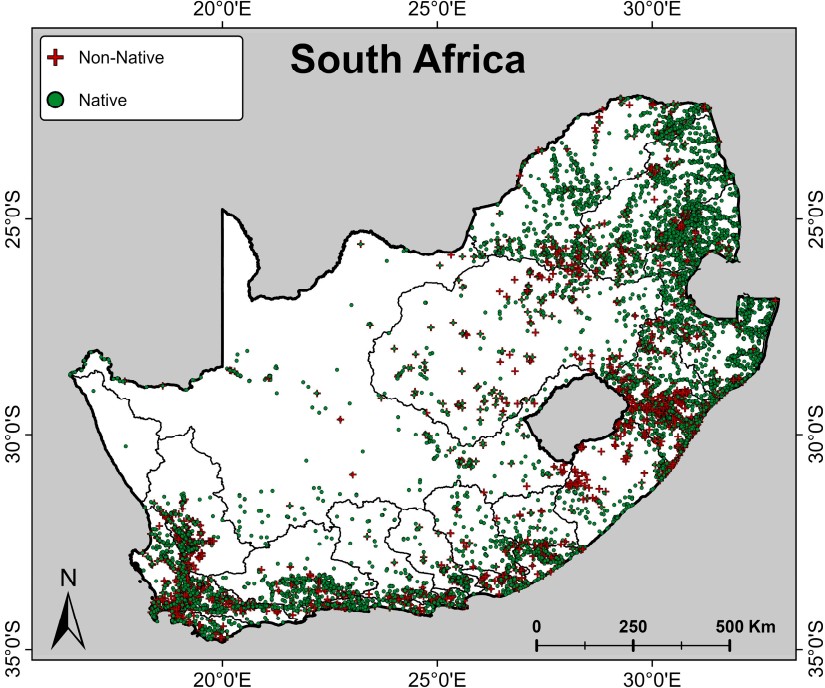

**Figure 2.** Republic of South Africa, showing primary catchment boundaries and the distribution of available, uncleaned freshwater fish occurrence records for native (green circles) and non-native (red crosses) species (*n* = 57,485 records).

### 3.1. Spatial Distribution of Records

The density of records per QDS grid cell shows that data were very unevenly distributed across the country (Figure 3). At the provincial level, fish occurrence records varied substantially between provinces (Table 1). Limpopo (*n* = 14,353), KwaZulu-Natal (*n* = 11,411), and Mpumalanga (*n* = 11,411) each had greater than 10,000 records, whilst Gauteng (*n* = 907) and Free State (*n* = 890) both had fewer than 1000 records, respectively. Similarly, the distribution of records among primary catchments varied widely (Table 2). Areas of relatively high density of occurrences (>100 records per QDC grid cell) were in the northeast (Limpopo, Olifants, Komati, Mfolozi, Tugela, and Mkomazi Primary Catchments) and southwest (Olifants/Doring, Berg, Breede, and Gourits Primary Catchments) of South Africa. Conversely, large areas with no records were observed within the central (Orange and Vaal Primary Catchments) and western (Buffels Primary Catchment) parts of the country. On a finer scale, there were noticeable gaps in data in the northern part of the Olifants Doring, Gourits, Mzimvubu, and Limpopo Primary Catchments, respectively. The vast majority of grid cells contained fewer (between 1–10) records (represented in violet; Figure 3).

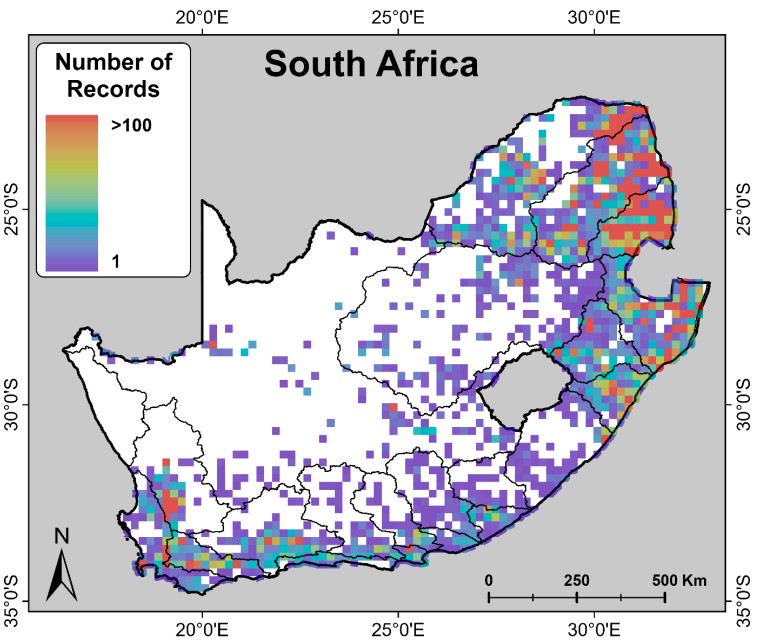

**Figure 3.** Density of native freshwater fish occurrence records across South Africa per quarter-degree square (QDS—15′) grid cell. Primary catchment boundaries are shown.

**Table 1.** Number of records and native, extralimital, non-native, endemic, and threatened species occurring in each province of South Africa.

| Province | Records | Native Species | Extralimital Species * | Non-Native Species | Endemic Species + | Threatened Species |
|---|---|---|---|---|---|---|
| Eastern Cape | 2725 | 35 | 10 | 14 | 13 | 8 |
| Free State | 890 | 19 | 4 | 7 | 8 | 0 |
| Gauteng | 907 | 38 | 3 | 9 | 5 | 1 |
| KwaZulu-Natal | 11,411 | 74 | 8 | 20 | 13 | 8 |
| Limpopo | 14,353 | 61 | 6 | 11 | 7 | 2 |
| Mpumalanga | 11,411 | 65 | 7 | 12 | 7 | 6 |
| North West | 1058 | 37 | 5 | 7 | 6 | 1 |
| Northern Cape | 1285 | 39 | 9 | 9 | 13 | 1 |
| Western Cape | 8159 | 30 | 10 | 16 | 19 | 11 |

\* As defined by Ellender and Weyl [32] and occurring outside of their home range. + Endemic refers only to Regional endemic level 2, Regional endemic level 1, Micro-endemic level 2, and Micro-endemic level 1 species, as defined by Dallas et al. [19]. Nationally endemic species were omitted.

**Table 2.** Number of records and native, extralimital, non-native, endemic, and threatened species occurring in each Primary Catchment of South Africa.

| Catchment ID | Catchment Name | Records | Native Species | Extralimital Species * | Non-Native Species | Endemic Species + | Threatened Species |
|---|---|---|---|---|---|---|---|
| Region A | Limpopo | 8355 | 60 | 7 | 9 | 7 | 1 |
| Region B | Olifants | 9823 | 59 | 7 | 10 | 6 | 4 |
| Region C | Vaal | 1240 | 34 | 6 | 10 | 7 | 1 |
| Region D | Orange | 1100 | 37 | 6 | 8 | 11 | 0 |
| Region E | Olifants/Doring | 3443 | 16 | 7 | 7 | 13 | 6 |
| Region F | Buffels | 6 | 0 | 3 | 0 | 0 | 0 |
| Region G | Berg | 1690 | 15 | 8 | 14 | 10 | 4 |
| Region H | Breede | 1902 | 13 | 5 | 10 | 10 | 3 |
| Region J | Gourits | 1269 | 12 | 7 | 7 | 7 | 2 |
| Region K | Kromme | 488 | 8 | 4 | 9 | 6 | 4 |
| Region L | Gamtoos | 433 | 10 | 4 | 6 | 6 | 2 |
| Region M | Swartkops | 309 | 10 | 3 | 6 | 4 | 1 |
| Region N | Sundays | 342 | 13 | 6 | 4 | 2 | 2 |
| Region P | Bushmans | 243 | 14 | 5 | 6 | 3 | 2 |
| Region Q | Fish | 293 | 11 | 5 | 7 | 2 | 2 |
| Region R | Keishkamma | 444 | 12 | 4 | 11 | 4 | 3 |
| Region S | Kei | 176 | 6 | 5 | 9 | 0 | 1 |
| Region T | Mzimvubu | 713 | 18 | 5 | 13 | 3 | 3 |
| Region U | Mkomazi | 2359 | 27 | 4 | 17 | 4 | 3 |
| Region V | Tugela | 2089 | 29 | 3 | 11 | 5 | 3 |
| Region W | Mfolozi | 7017 | 68 | 6 | 12 | 8 | 6 |
| Region X | Komati | 8471 | 58 | 3 | 12 | 4 | 5 |

* As defined by Ellender and Weyl [32] and occurring outside of their home range. + Endemic refers only to Regional endemic level 2, Regional endemic level 1, Micro-endemic level 2, and Micro-endemic level 1 species. Nationally endemic species were omitted.

### 3.2. Species Richness

Native species richness per QDS grid cell (Figure 4A) followed a similar pattern to the record density per grid cell (Figure 3) across South Africa. There were noticeable areas of high species richness in Mpumalanga, Limpopo, and northern KwaZulu-Natal (corresponding with the Limpopo, Olifants, Komati, and Mfolozi Primary Catchments). Additionally, species richness was relatively high (between 10–15 species per QDS grid cell) along the east coast of KwaZulu-Natal (Tugela and Mkomazi Primary Catchments) and in a small cluster in the Western Cape (along the Berg-Olfants/Doring Primary Catchment boundary). As expected, based on the recorded occurrence density (Figure 3), species richness was lowest in the Free State, Northern Cape, and North West Province (Orange, Vaal and Buffles Primary Catchments).

Endemic species richness per QDS grid cell (Figure 4B) was highest in the Western Cape Province (Olifants/Doring, Berg, and Bree Primary Catchments) and along the south coast of the Eastern Cape (Swartkops, Kromme, and Gamtoos Primary Catchments). There were small clusters of higher endemic species richness in the northeastern part of the country (Komati, Mfolozi, and Vaal Primary Catchments).

Threatened species were distributed fairly consistently in Limpopo and Mpumalanga (Limpopo, Olifants, and Komati Primary Catchments), along the east coast of KwaZulu-Natal (Mfolozi, Tugela, and Mkomazi Primary Catchments), and south coast of the Eastern Cape (Keishkamma, Bushmans, Swartkops, Fish, Sundays, Gamtoos, Kromme, and Gourits Primary Catchments) (Figure 4C). The Western Cape (Olifants/Doring, Berg, and Bree Primary Catchments) contained the highest concentration of threatened species, followed by the Komati Primary Catchment in the KwaZulu-Natal. There were also notable hotspots within the Kromme and Keishkamma Primary Catchments, as well as within the Mkomazi and Mfolozi Primary Catchments (Figure 4C).

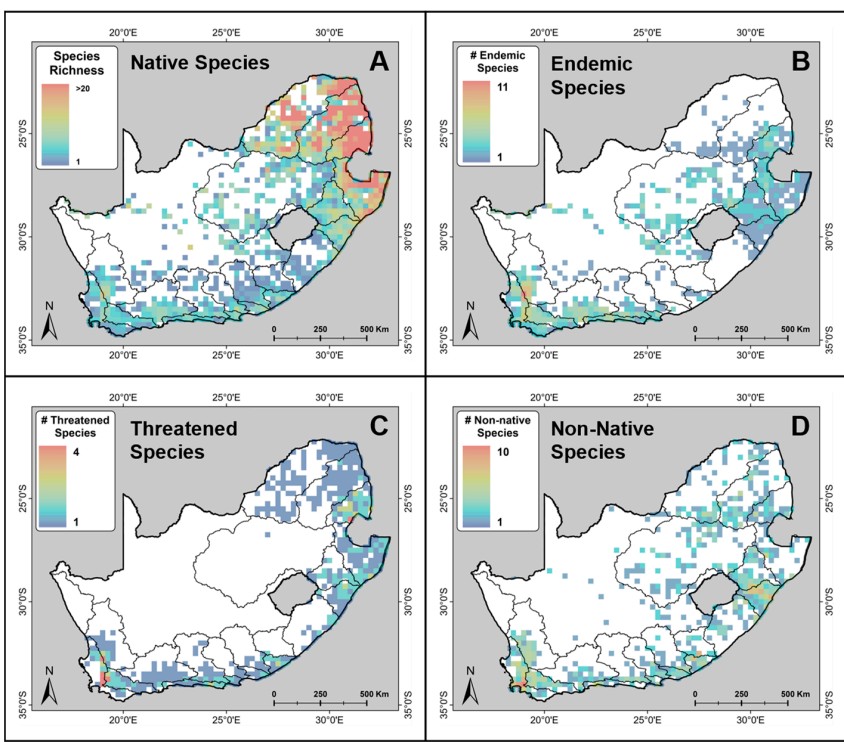

**Figure 4.** Species richness per quarter-degree square (QDS—15′) grid cell for all native (**A**), endemic* (**B**), threatened+ (**C**), and non-native (**D**) freshwater water fish species occurring in South Africa. Primary catchment boundaries are shown. Endemic refers only to Regional endemic level 2, Regional endemic level 1, Micro-endemic level 2, and Micro-endemic level 1 species, as defined by Dallas et al. [19] Nationally endemic species were omitted. Threatened species refers to freshwater water fish listed as threatened according to the IUCN Red List of Threatened Species (Vulnerable, Endangered, or Critically Endangered).

Non-native species richness per QDS grid cell (Figure 4D) was relatively high (4–7 species) along the east coast of KwaZulu-Natal (Mkomazi and Keishkamma Primary Catchments) and highest in the Western Cape (Olifants/Doring, Berg, and Bree Primary Catchments). In general, non-native species were present in all provinces and catchments where records were available (Figure 4D). Importantly, 'species richness' hotspots for non-native species indicate grid cells where several non-native species have established successful, self-sustaining (often invasive) populations.

*3.3. Protected Areas*

Of the 47,946 records for native fish species analysed in this study, 47% (*n* = 22,756) occurred outside of protected areas, whilst 53% (*n* = 25,190) were located inside protected areas (Figure 5). Of these, 28% (*n* = 6984) were located within a formally protected area, whereas the remaining 72% (*n* = 18,206) were located within a conservation area (Figure 5), as defined by the South Africa Protected and Conservation Areas Database [35,36]. When assessing records for threatened species, a total of 5740 records were used in the final analyses. Of these, 43% (*n* = 2464) were located outside a protected area, and 57% (*n* = 3276) within either a formally protected area (32%; *n* = 1060) or a conservation area (68%; *n* = 2216) (Figure 6A). Based on the QDS grid cell combined distribution, threatened species covered an area of 328,319 km², with only 36% of this range overlapping with South Africa's protected area network (Figure 6B). However, all QDS grid cells that contained more than three threatened species (i.e., the highest density of threatened species) overlapped with a protected area (Figure 6C). Concerningly, 58% of threatened species co-occurred in the same grid cells as non-native species (Figure 6D).

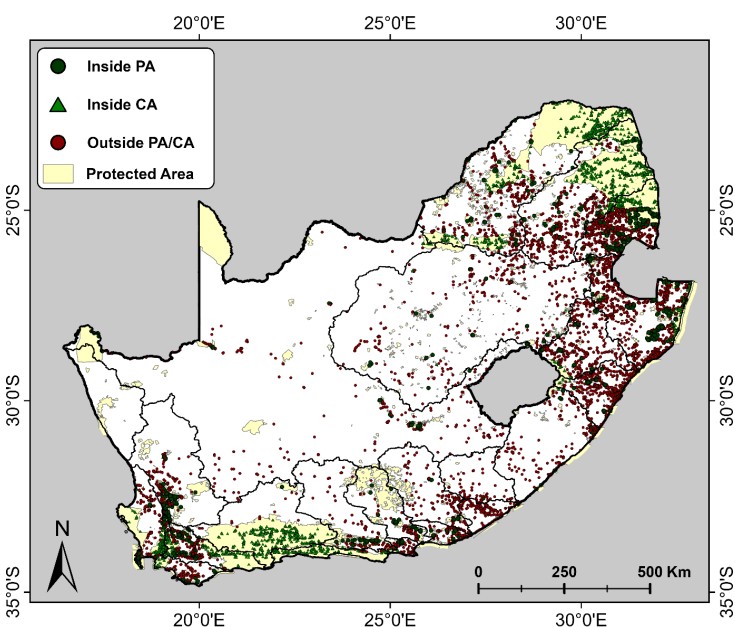

**Figure 5.** Freshwater fish occurrence records for native freshwater fish occurring within protected areas (green circles), conservation areas (green triangles), and outside of either protected or conservation areas (red circles) in South Africa. Primary catchment boundaries are shown.

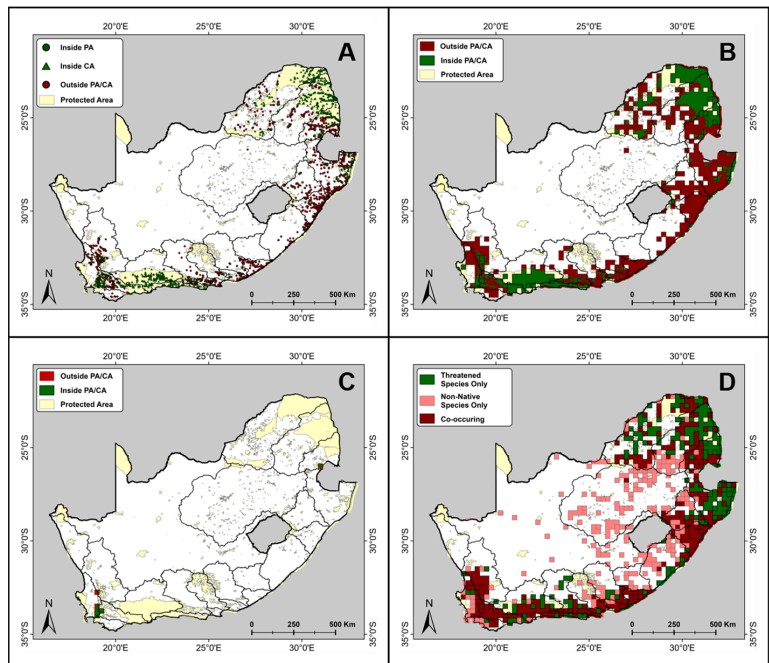

**Figure 6.** Primary Catchment map of the Republic of South Africa, showing the distribution of threatened freshwater fish (listed as either Vulnerable, Endangered, or Critically Endangered according to the IUCN Red List of Threatened Species) within South Africa's protected areas network. Panel (**A**) shows threatened occurrence records within protected areas (green circles), conservation areas (green triangles), and outside of either protected or conservation areas (red circles). Panel (**B**) shows QDS grid cell distribution of threatened freshwater fish occurring within protected or conservation areas (green) and outside of protected or conservation areas (red). Panel (**C**) shows QDS grid cells that contain the most (*n* = 4) threatened species. Panel (**D**) shows QDS grid cell co-distributions for threatened and non-native species, indicating where threatened species do not overlap with non-native species (green), where threatened species overlap with non-native species (red), and where non-native species do not overlap with threatened species (pink).

A total of 4244 records for non-native species were used in the final analyses for non-native species occurring in South Africa. Of these 62% (*n*= 2625) were recorded outside a formally protected area, whilst 38% (*n*= 1619) were recorded inside either a protected area (53%; *n* = 870) or a conservation area (47%; *n* = 749) (Figure 7A). South Africa's total terrestrial protected area network covers 272,485 km$^2$, of which 100,815 km$^2$ (37%) overlaps with non-native species distributions based on species richness grids generated at quarter-degree square (QDS—15') resolution (Figure 7B).

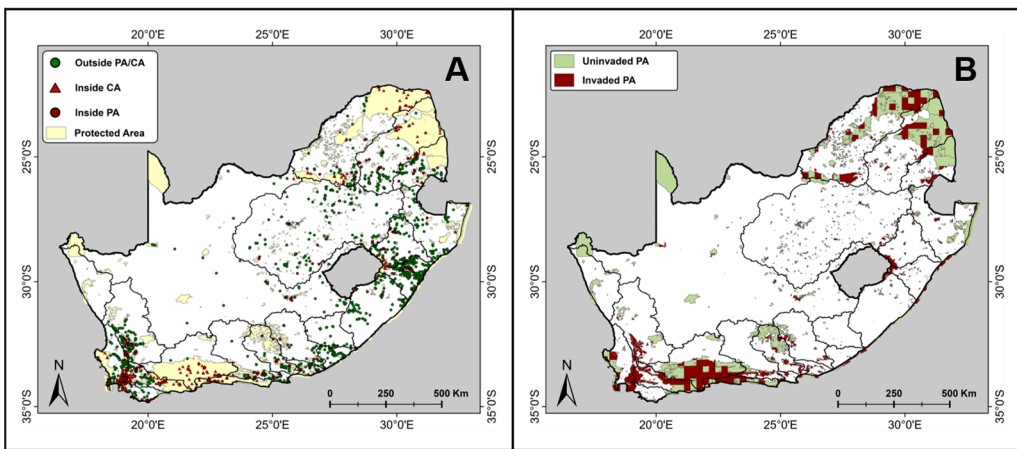

**Figure 7.** Primary Catchment map of the Republic of South Africa showing (**A**) records for non-native freshwater fish occurring within protected areas (red circles), conservation areas (red triangles), and outside of either protected or conservation areas (green circles); and (**B**) QDS grid cell distribution of non-native freshwater fish within protected or conservation areas (red).

## 4. Discussion

We found that record density varied spatially, at both primary catchment and provincial scales. The diversity of freshwater fishes also varied spatially. Native fish hotspots were identified in Limpopo, Mpumalanga, and KwaZulu-Natal; an endemic species hotspot was identified in the Western Cape, and threatened species hotspots were identified in the Western Cape, Mpumalanga, Eastern Cape, and KwaZulu-Natal. Non-native species hotspots mirrored threatened species hotspots in the Western Cape, Mpumalanga, Eastern Cape, and KwaZulu-Natal. A total of 43% of threatened species records fell outside of protected areas, and 38% of non-native species records fell within protected areas.

### 4.1. Spatial Distribution of Records

The spatial distribution of freshwater fish records was uneven across South Africa, indicating that sampling across the country has been biased and concentrated in a few regions. There are several factors that can influence the distribution of these species occurrence data. These include the spatial extent of the original study [56,57], how data from individual studies were stored [56], accessibility of sites [58], and proximity to man-made infrastructure such as roads [58,59], cities [60], and research institutions [59]. There is also evidence that a higher proportion of occurrence records are collected within protected areas, or hotspots of species richness [61,62]. Whilst there have been numerous studies [59,60,63–67] that have sought to identify and quantify inherent bias when using large, publicly available species occurrence data sets, adequately dealing with these biases has proven difficult, and in some cases, even impossible [63]. One concern with working with large, historic data sets is how these underlying biases influence analyses and ultimately the conclusions that can be drawn. Large species occurrence data sets that are prone to bias can have far-reaching implications for the perception of, and inferences about, macroecological patterns [66], which limits the usefulness of these outputs. Another major issue with large historic data sets is that species absences are very rarely reported [68]. This obviously adds additional

uncertainty when interpreting analyses based on these data. For example, a data gap could mean that: (i) a site was not sampled; (ii) a site was sampled and did not contain any fish species; or (iii) a site was sampled but there was a failure to detect all fish species present. This is especially the case for rare species, i.e., those likely to be represented by only one or two individuals per site [69,70]. In this regard, interpretation of results from large, historic data sets should be approached with caution.

Whilst the scope of this study did not include assessing the level of bias and spatial autocorrelation between the distribution of man-made landmarks and the FBIS fish data set, this is an important next step that should be carried out when conducting future analyses at national, provincial, or catchment scales. Quantifying this will allow for better contextualisation of these results. Additionally, the unevenness in available freshwater fish data in South Africa speaks to the urgent need for a coordinated national and provincial fish monitoring programme. Unfortunately, however, such biodiversity data depend on a wealth of scientific, human, and financial resources [65], which can be a limiting factor in South Africa's current socio-political and economic landscape [71]. However, given the importance of comprehensive and up-to-date biodiversity data for making informed conservation and management decisions, adequately and strategically monitoring river systems across the country should be prioritised.

Additionally, it is important to acknowledge that using a QDS resolution for this assessment provided a useful, but generalised, insight into freshwater fish spatial patterns in South Africa at a national scale. River habitats are linear systems that are not equally distributed across the geographic landscape. As such, many QDS grid cells are simply not sampled because there are no surface water or drainage regions with regular enough surface water to sustain permanent fish populations.

### 4.2. Species Richness

The patterns of species richness, endemism, and threatened species are congruent with previously published descriptions of South Africa's freshwater fish diversity. Primary catchments containing the highest freshwater fish species richness in South Africa included the Limpopo, Olifants, Komati, Mfolozi, Tugela, and Mkomazi Primary Catchments, as well as along the Berg-Olfiants/Doring Primary Catchment boundary. Skelton [15] also identified richness hotspots in the northeastern region of the country, with endemism and threat hotspots in the Western and Eastern Cape [15]. Interestingly, Skelton [15] attributed these hotspots to the relatively high topographical relief characteristic of the Cape Fold Mountains. This hypothesis was supported by more recent work [72,73], which found that South Africa's complex geological and climate history, characterised by tectonism and sea-level fluctuations, created unique biogeographic conditions that allowed for the diversification of stream-dwelling taxa, particularly obligate freshwater fishes [72,74]. More recently, Chakona et al. [13] assessed the distribution of freshwater fishes in South Africa, using distribution records from SAIAB's National Fish Collection only (a subset representing ~40% of the raw data included in our study). Whilst they assessed richness at the ecoregion scale, broad patterns in species richness are similar to those presented here. Consequently, it is safe to conclude that the broad patterns of richness identified are an accurate representation of the current state of freshwater fishes in South Africa—despite the concerns regarding the bias contained within the currently available data set. Similarly, endemic species hotspots were found in the Olfiants/Doring, Berg, and Bree Primary Catchments, and, to a lesser degree, the Swartkops, Kromme, Gamtoos, Komati, Mfolozi, and Vaal Primary Catchments. Threatened species hotspots were identified in the Olifants/Doring, Berg, and Bree Primary Catchments, as well as the Mkomazi and Mfolozi catchments, respectively. It is thus recommended that all QDS grid cells identified as having high levels of species richness, endemism, and threatened species be prioritised for resampling and monitoring, to better inform the conservation interventions required in these catchments. Focussing effort and resources in this targeted manner could

provide the most efficient use of the limited national, provincial, and scientific resources available in the country.

Of particular concern was the large overlap between the distributions of threatened species and non-native species in South Africa, with the majority (58%) of threatened species co-occurring with non-native species (Figure 6D). Non-native species can have profound and devastating impacts on both native threatened species and, more broadly, freshwater habitats as a whole, and are considered a top threat to native freshwater fish in South Africa [13,32–34,75,76]. The presence of non-native species has resulted in the widescale extirpation of many native species from their historic distributions (especially in mainstem rivers), with many native species now relegated to the upper reaches of tributaries, surviving in small, fragmented populations above waterfalls or other physical barriers that have prevented invasion by non-native species [32,34,77].

It is also important to note that this study was limited to formally described primary and secondary freshwater, as well as catadromous, fishes based on the GBIF taxonomic backbone and available IUCN Red List assessments. As such, the analyses presented will likely need to be updated in the near future once: (i) experts settle on an updated species list for freshwater fishes in the country; (ii) all species in South Africa have had their threat status assessed; and (iii) ongoing taxonomic revisions for several species suites are formalised [13,78–82].

### 4.3. Protected Areas

We found that South Africa's protected area network [35,36] does not adequately cover the distributions of threatened species in the country. That 57% of South Africa's freshwater fish records were located within a protected or conservation area suggests a strong bias towards sampling in those areas, because they account for only ~20% of South Africa's total land surface area (Figure 6A). Moreover, only 36% of the total area occupied by threatened species occurs within a protected or conservation area (Figure 6B). On the one hand, this indicates that the majority of threatened species distributions are not under formal protection and at heightened risk of extinction. Conversely, however, this level of protection is relatively high, when compared to the percentage of terrestrial land area under protection globally [38,49]. Furthermore, 37% of the country's formal protected area network is invaded by non-native species, with a high percentage over-lap between threatened and non-native species at the QDS scale (Figure 6D). As such, threatened freshwater fish in South Africa still face direct threats from non-native species, even though it may appear that these species are well-considered in the protected area network. Thus, our findings add further evidence to the growing body of research that considers South Africa's protected area network to provide inadequate protection for sensitive freshwater species. Kleynhans [83], Nel et al. [43], Abell et al. [44], and Nel at al. [49] all concluded that South Africa's protected areas did not adequately conserve freshwater ecosystems, with the majority of rivers falling outside the protected area network. Of the rivers located in South Africa, 70% are classified as either Not Protected or Poorly Protected [39]. Furthermore, of the river systems that are formally protected, almost half of these have already been degraded by human activities upstream of the protected area [49,83].

For example, a comprehensive assessment of the 19 National Parks managed by SAN-Parks found that the National Parks protected network only includes small components of protected river areas that form part of much larger, degraded aquatic systems further up- and downstream of the parks [45]. Consequently, very few sites within National Parks contain freshwater fishes that are not under direct threat from land-use change, habitat loss, and non-native species impacts [45]. Similarly, an analysis of the National Parks and nature reserves within the Cape Floral Kingdom (roughly the same geographic range as the Western Cape Province) found that, whilst protected areas do contain populations of most native fish, actual protection was impaired because species ranges extended beyond the boundaries of protection or were protected in rivers with substantial invasion by non-native

species [44,46]. More recently, Jordaan et al. [34] also found that protected areas in the Cape Fold Ecoregion did not adequately protect native freshwater fish, with 84% of taxa regarded as under-protected [34].

However, there has been some improvement to the level of protection afforded to South Africa's threatened fish populations. Firstly, the NFEPA project developed a series of strategic spatial maps, prioritising the conservation of the country's freshwater ecosystems [16]. Importantly, the NFEPA provides for Fish Sanctuaries and associated Fish Support Areas, which includes rivers that are essential for protecting threatened and near-threatened freshwater fish native to South Africa [16]. Furthermore, the NFEPA also highlights important Upstream Management Areas, which flag sub-quaternary catchments where human activities need to be carefully managed to prevent degradation of downstream river Fish Sanctuaries and Fish Support Areas [16]. More recently, Kajee et al. [82] reported on the first inclusion of freshwater fishes into the DFFE National Environmental Screening Tool [84]. This process provided an additional layer of protection for South Africa's threatened freshwater fish species, along roughly 50,000 km of river habitat [82]. However, it is also important to acknowledge that South Africa's freshwater fish fauna have the potential to serve as a vital food and income source for rural communities that face extreme levels of poverty and food insecurity [85]. Subsistence fishing activity, in response to modern socio-economic circumstances, was recorded at 77% of dams in South Africa, with recent studies indicating that more than 1.5 million people are involved in freshwater angling activities, in an industry worth approximately ZAR 9 billion annually [85,86]. Consequently, there is a need to reimagine the country's protected area network to better safeguard freshwater fish, and freshwater habitats in general, whilst also accounting for the socio-economic needs of rural communities in South Africa.

*4.4. Limitations*

Whilst these results will no doubt be useful for researchers, catchment and provincial conservation managers, and policymakers, there are limitations to our study. Given that the basis of this assessment is occurrence records sourced from an open access biodiversity database [19,51], analyses at national (and even sub-national) scales were complicated by a lack of consistency in the types of metadata available for each record. For example, inconsistent abundance and effort data limited our ability to undertake additional diversity analyses. Additionally, working with a large data set (in our case, more than 50000 records) can make manual cleaning of raw data unsustainably time-consuming. In this study, we chose to follow the methods of Robertson et al. [55] and used the biogeo R package to 'autoclean' our data prior to analysis. Whilst this approach seems to have worked well when assessing species richness patterns across the country, further scrutiny of species lists at provincial and catchment levels revealed some inaccuracies.

For example, when reviewing the final species lists for the Olifants/Doring Primary Catchment (a catchment known to the authors), we found that there were inaccuracies in the final number of native, endemic, and threatened species reported in the catchment. Most notably, our data indicated that there were 16 native species occurring in the catchment. However, a literature search, along with expert consultations, revealed 11 such species in this catchment. Further scrutiny revealed that five species (*Labeo rosae*, *Pseudobarbus burchelli*, *Pseudobarbus burgi*, *Pseudobarbus capensis*, and *Sandelia capensis*) had occurrence records in the catchment, when in fact none of these species are known to occur in the area. These records ($n < 10$) are likely a result of misidentifications or inaccurate GPS coordinates. Regardless, their existence undermines the potential benefits of using large, historic data sets to assess species diversity at large spatial extents. Accurate cleaning of this dataset prior to analysis was beyond the scope of this study but should be a priority for such analyses in the future. This would need to be expert-driven and would require substantial time and resources. We suggest that such a project be planned and coordinated by a national body, such as the SANBI or the SAIAB. As such, we chose to include these occurrences in our assessment, but to flag them in our final species list reports (see Supplementary Materials S2 and S3).

Additionally, our study focussed only on the spatial patterns of freshwater fishes, without conducting a detailed temporal analysis of the data, which fell outside the scope of our study. It is important to acknowledge that these data could, and likely are, influenced by inter-specific temporal variation in the data. Consequently, this study, and the published species list, should act only as a first step towards understanding species richness at national, provincial, and catchment scales. A logical next step would be to repeat this study, after having spent the time to thoroughly clean the data set (as was done for threatened fish taxa only by Kajee et al. [82]) and conduct a comprehensive temporal analysis of the data.

## 5. Conclusions

We present the first assessment of the status of freshwater fish distributions in South Africa using all available data sources from the FBIS. While acknowledging the shortcomings of working with a large, historic data set is important, the patterns emerging from this assessment do adequately identify key fish richness, endemism, and threat hotspots. These patterns are broadly aligned with historic and current expert-driven assessments. This provides a much-needed snapshot of the most important geographic areas for freshwater fishes, and a valuable resource for identifying scientific, conservation, and management priorities in South Africa. We also present the first national-scale assessment of the effectiveness (in terms of geographic coverage and invasion status) of South Africa's protected area network in protecting threatened freshwater fishes. We concluded that the current protected area network is not sufficient to functionally conserve threatened species and prevent future population or species extinctions, given that the majority of the distributions of these species are either outside of the protected area network, invaded by non-native species, or do not have adequate upstream protection. Future interventions should prioritise systematically sampling river systems to fill in identified data gaps, developing strategic long-term monitoring programmes in key hotspot catchments, the comprehensive cleaning of available data to produce accurate distribution maps for all species, and reimagining protected areas to better conserve freshwater fishes.

**Supplementary Materials:** The following supporting information can be downloaded at: https://www.mdpi.com/article/10.3390/fishes8120571/s1, Supplementary Material S1: List of native species that have been translocated outside of their native range, extracted from Ellender and Weyl [32].; Supplementary Material S2: List of all native freshwater fishes occuring in each province in South Africa; Supplementary Material S3: List of all native freshwater fishes occuring in each primary catchment in South Africa. Supplementary Material S4: List of data sources on which the analyses presented in this paper are based.

**Author Contributions:** Conceptualization, M.K., H.F.D., C.L.G., C.J.K. and J.M.S.; methodology, M.K. and J.M.S.; software, M.K.; validation, M.K., H.F.D., J.M.S. and C.J.K.; formal analysis, M.K.; investigation, M.K.; resources, H.F.D., J.M.S. and C.L.G.; data curation, M.K., H.F.D. and J.M.S.; writing—original draft preparation, M.K. and J.M.S.; writing—review and editing, M.K., H.F.D., C.L.G., C.J.K. and J.M.S.; visualization, M.K.; supervision, H.F.D., C.L.G. and J.M.S.; project administration, C.L.G.; funding acquisition, H.F.D. and J.M.S. All authors have read and agreed to the published version of the manuscript.

**Funding:** This research was funded by the JRS Biodiversity Foundation, grants 60606 and 60919, and the South African National Biodiversity Institute (SANBI). This work is based on research supported in part by the National Research Foundation (NRF) of South Africa, grant number MND2000621534710–UID: 133692, and the NRF-SAIAB DSI/NRF Research Chair in Inland Fisheries and Freshwater Ecology (UID 110507). Student funding, in the form of an MSc bursary, was also provided by the Freshwater Biodiversity Unit of the South African National Biodiversity Institute (SANBI).

**Institutional Review Board Statement:** All the data come from previous research or other databases. In this case, ethical approval is not needed for this article.

**Data Availability Statement:** The raw data presented in this study are openly available on the Freshwater Biodiversity Information System (FBIS) at https://freshwaterbiodiversity.org/ The data set can also be accessed via GBIF at Freshwater Biodiversity Information System (FBIS) Fish Data. Version 1.6. Freshwater Research Centre [https://doi.org/10.15468/gmk6hg]. All additional data and analyses are available on request from the corresponding author.

**Acknowledgments:** We would like to thank the FRC staff and interns, specifically Aneri Swanepoel, who provided invaluable assistance with data collection and cleaning. We express our gratitude to the Kartoza team for assistance with the technical development and implementation of the FBIS platform. Lastly, we would like to acknowledge the contributions of Dominic Henry, Res Altwegg, and Vernon Visser, who all provided invaluable advice with aspects of the data analyses. The authors acknowledge the individuals and organisations whose data have been used in the analyses presented in this paper.

**Conflicts of Interest:** The authors declare no conflict of interest.

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
