# Peer review of "The Status of South Africa’s Freshwater Fish Fauna: A Spatial Analysis of Diversity, Threat, Invasion, and Protection"

_fishes, doi:10.3390/fishes8120571_

Round 1
Reviewer 1 Report
Comments and Suggestions for Authors RE: The status of South Africa’s freshwater fish fauna: A spatial analysis of diversity, threat, invasion, and protection
This MS synthesises a large database with relation to protected area, endemicity and invasion status of South African freshwaters. The information is needed for policy and legislation development, and is one of the few locations where this sort of data is synthesised across such a large area. Minor edits needed.
Introduction
Generally very thorough and well communicated rationale and background, especially the historical trends in data accessibility and the role of national organisations. A few sentences, besides those in the final intro paragraph, characterising some more specific stressors (beside habitat loss) in SA freshwaters would help strengthen the case though in a more general way (rather than a touting of FBIS).
Check your wording, L169 – still has “this chapter” – ie straight from the thesis.
Methods
It would be helpful for the international community to also have ecoregions specified and shown in the maps in methods. As well as a general overview of the different types of systems in SA – rivers, reservoirs, geomorphological characteristics – this can link into the ecoregion aspect and will help contextualise the results.
L194 – Weyl et al. 2020 book chapter is the more updated list – why was this not used aswell?
L198 – some explanation of what autoclean is, is needed here
Results
L226 – change south African to south Africa
Is fig 3 the density of records – ie number of records for a QDS or the species richness of the QDS? This needs to be clarified as it seems to be number of records, but text suggests richness
Check your figure captions, there are two fig 3s. This is causing the confusion in the comment above
I think that some comment on the temporal aspect of the dataset is needed in both the results and discussion, as trends in reporting aswell as change in distributions will be reflected, consider adding some aspect in to the results for context.
Discussion
L411 – some further contextual information re river capture and topography would be useful as the paleogeography of the region is crucial to the species richness patterns
L438 – not sure about the phrase “small refugee populations” …
L496 – reimaging or reimagining?
Somewhere in the discussion, besides conservation, there needs to be a nod to livelihood contribution of inland fisheries in SA – conservation and subsistence fisheries go somewhat hand in hand – especially in the context of south Africa and food poverty, the stakeholders need to be acknowledged
Author Response
Dear Reviewer 1,
Thank you very much for your careful and thoughtful review of our manuscript. We appreciate your valuable feedback and suggestions, which have helped us to improve our paper significantly.
We have carefully considered all of your comments and have addressed them in detail in our point-by-point response. We believe that our revisions have addressed all of your concerns and have strengthened the overall quality of the manuscript.
We are grateful for your time and effort in reviewing our manuscript. Your feedback has been invaluable to us.
Sincerely
Please see the attachment.

Reviewer 2 Report
Comments and Suggestions for Authors
This study evaluates freshwater fish diversity, threat, invasion, and protection status in South Africa based on database of Freshwater Biodiversity Information System (FBIS). It possibly presents a case of country-scale assessment of the protecting effectiveness to threatened freshwater fishes, and warns that the current protected area network does not work well to conserve threatened species and prevent extinctions. It is worth to be published after careful revisions.
Comments to Authors:
1. Title-It will be better to change the former Title to “The status assessment of South Africa’s freshwater fish fauna using database of Freshwater Biodiversity Information System”.
2. Abstract-(1) Give the exact meaning for “Western Cape”, “Mpumalanga”, “Eastern Cape”, “KwaZulu-Natal”. Are they names of river, dam, freshwater lake habitats? (2). Replace “invaded by non-native species” to “devastatingly impacted by non-native species”.
3. Introduction-(1) The authors should focus on research and monitoring advance in recent decades about freshwater fish diversity, threat, invasion, and protection status in South Africa. The history mention of freshwater fish research dating back 100-200 is not needed. (2) This chapter is too long. One half of the content needs to be drastically cut drastically reduced. (3) Replacing “This chapter aims to” by “This paper aims to”.
4. Materials and Methods-(1) 2.1. Study Site- What the exact meaning for “A = Limpopo; B = Olifants; C = Vaal; D = Orange; E = Olifants/Doring; F = Buffels; G = Berg; H = Breede; J = Gourits; K = Kromme; L = Gamtoos; M = Swartkops; N = Sundays; P = Bushmans; Q = Fish; R = Keishkamma; S = Kei; T = Mzimvubu; U =Mkomazi; V = Tugela; W = Mfolozi; and X = Komati”? Are they names of river, dam, freshwater lake habitats, or those for special locations?
5. 3. Results-No actual results can be found in this paper for primary/secondary freshwater fish, and for catadromous species. Not for diadromous species? No anadromous species in South Africa?
6. 4. Discussion-Please discuss more on limitations of Freshwater Biodiversity Information System (FBIS) and possible improving strategies for next step research.
Author Response
Dear Reviewer 2,
We would like to thank you for your insightful and constructive review of our manuscript. We appreciate your careful reading of our work and your valuable suggestions for improvement. We have considered all of your comments and have addressed them in detail in our point-by-point response. We have also provided a rationale for any instances where we have deviated from your suggestions.
We would like to particularly thank you for your feedback on the stylistic aspects of our manuscript. We have carefully considered your suggestions and have made revisions to our writing in accordance with your recommendations. However, in some cases, we have respectfully disagreed with your suggestions. In these instances, we have provided a rationale for our decision. We believe that our revisions have resulted in a more polished and professional manuscript.
Thank you again for your time and effort in reviewing our manuscript. We appreciate your willingness to help us improve our work.
Sincerely
Please see the attachment.

Reviewer 3 Report
Comments and Suggestions for Authors
Dear Authors,
The submission entitled “The status of South Africa’s freshwater fish fauna: A spatial analysis of diversity, threat, invasion, and protection”. aimed to use FBIS database to assess spatial patterns of distribution, diversity, invasion and threat status in South Africa at provincial and primary catchment scales. The manuscript is written well. This is really very interesting study and I strongly recommend it as a “review” after revising the manuscript slightly. Please, see the minor revisions attached.
Kind regards

Author Response
Dear Reviewer 3,
We would like to express our sincere gratitude for your insightful and constructive review of our manuscript. We value your careful consideration of our work and your valuable suggestions for improvement. We have meticulously reviewed all of your comments and have addressed them in detail in our point-by-point response.
We are particularly appreciative of your feedback on the clarity and conciseness of our writing. We have carefully considered your suggestions and have revised our manuscript to improve its readability and flow. We believe that your feedback has significantly enhanced the overall quality of our paper.
Thank you again for your time and effort in reviewing our manuscript. We are grateful for your willingness to share your expertise and help us improve our work.
Sincerely
Please see the attachment.
